# Abundance and viability of particle-attached and free-floating bacteria in dusty and nondust air

Wei Hu[1,2], Kotaro Murata[2,3], Chunlan Fan[2], Shu Huang[1], Hiromi Matsusaki[2], Pingqing Fu[1], Daizhou Zhang[2]

[1] Institute of Surface-Earth System Science, School of Earth System Science, Tianjin University, Tianjin, 300072, China
[2] Faculty of Environmental and Symbiotic Sciences, Prefectural University of Kumamoto, Kumamoto, 862-8502, Japan
[3] Department of Physics, Tokyo Gakugei University, Tokyo, 184-8501, Japan

*Correspondence to*: Daizhou Zhang (dzzhang@pu-kumamoto.ac.jp)

**Abstract.** Airborne bacteria are widespread as a major proportion of bioaerosols and their coexistence with dust particles enables both bacteria and dust particles to be more active in ice cloud formation and to be harmful to public health. However, the abundance and viability of particle-attached and free-floating bacteria in dusty air have not been quantitatively investigated. We researched this subject based on the fact that airborne bacterial cells are approximately 1 µm or smaller in aerodynamic diameter; therefore, particle-attached bacteria should occur in aerosol samples of particles larger than 1 µm, and free-floating bacteria should occur among particles smaller than 1 µm. Our observations at a coastal site in Japan in spring, when the westerlies frequently transported dust from the Asian continent, revealed that particle-attached bacteria in dust episodes, at the concentration of $3.2 \pm 2.1 \times 10^5$ cells m$^{-3}$ on average, occupied $72 \pm 9$ % of the total bacteria. In contrast, the fraction was $56 \pm 17$ % during nondust periods and the concentration was $1.1 \pm 0.7 \times 10^5$ cells m$^{-3}$. The viability, defined as the ratio of viable cells to total cells, of particle-attached bacteria was $69 \pm 19$ % in dust episodes and $60 \pm 22$ % during nondust periods on average, both of which were considerably lower than the viabilities of free-floating bacteria (about 87 %) under either dusty or nondust conditions. The presented cases suggest that dust particles carried substantial amounts of bacteria on their surfaces, more than half of which were viable, and spread these bacteria through the atmosphere. This implies that dust and bacteria have important roles as internally mixed assemblages in cloud formation and in linking geographically isolated microbial communities, as well as possibly have synergistic impact on human health.

## 1 Introduction

Biological particles in the atmosphere have a potentially significant effect on climate change (Ariya et al., 2009;Delort et al., 2010;Möhler et al., 2007;Zhang et al., 2017), efficiently link microbial communities between continents, islands and oceans (Fröhlich-Nowoisky et al., 2016;Morris et al., 2011;Caliz et al., 2018), and pose risks to human health (Polymenakou et al., 2008;Reinmuth-Selzle et al., 2017). Representing a high fraction of primary biological particles, airborne bacteria are emitted into the atmosphere from various sources, among which desert dust is a major source (Morris et al., 2011;Pöschl and Shiraiwa, 2015;Pöschl et al., 2010). The cooccurrence of dust and high concentrations of bacteria has been observed frequently in different locations, indicating the widespread nature and dissemination of bacteria with dust at local, regional and even global scales (Griffin, 2007;Hara and Zhang, 2012;Iwasaka et al., 2009). Limited available observations have revealed the coexistence of mineral and biological contents in ice crystals (Creamean et al., 2013;Pratt et al., 2009), and laboratory experiments have demonstrated that the ice nucleation ability of dust particles is enhanced by biological components, including bacteria in the particles (Boose et al., 2019;Tobo et al., 2019;Conen et al., 2011). Recent toxicological studies with mouse exposure found that the internal mixture of dust and pathogenic bacteria exacerbated pneumonia (He et al., 2012). In addition, the attachment of bacteria to dust particles is expected to largely alter the fate of bacterial cells in the air due to protection by the dust particles from harsh environmental conditions (Bowers et al., 2013) and enhanced gravitational settling (Zhang, 2008). All these results reflect that the adherence of bacterial cells to dust particles, i.e., the particle-attached state, and the viability or metabolic capability of bacterial cells are key factors affecting the roles and fate of airborne bacteria in the evolution, development and conservation of the natural environment.

Quantitative data on the mutual state of airborne bacteria and dust particles in dusty air are without any doubt scientifically very interesting (Schuerger et al., 2018), but are rare because of a lack of available and confident methods, leaving unidentifiable uncertainties in both field observations and model simulations exploring the activities and roles of bacterial cells in atmospheric processes. The cell size distributions for bacteria separated from soils have previously been investigated (Portillo et al., 2013). Whereas, the survival strategies, dispersal processes and size distribution of airborne bacteria should be different from those of bacteria in soils. The possible causes are that the aerosolization efficiency of soil bacteria from Earth surfaces varies according to bacterial species and soil types (Joung et al., 2017) and airborne bacteria suffer air turbulence and harsh atmospheric stressors (Hara and Zhang, 2012). Bacteria-associated particles in the air have an aerodynamic diameter significantly larger than the typical size (approximately 1 μm) of individual bacterial cells (Burrows et al., 2009). This is because airborne bacterial cells are favorably attached to coarse particles, such as dust particles and plant debris, or are sometimes found as assemblages of many cells (Després et al., 2012;Iwasaka et al., 2009;Maki et al., 2013;Lighthart, 1997). We quantified the fractions of particle-attached and free-floating bacterial cells in dusty and nondust air based on the fact that airborne bacterial cells are usually ~1 μm or smaller than 1 μm (Delort et al., 2010;Després et al., 2012;Pósfai et al., 2003;Burrows et al., 2009;Hara et al., 2011); thus, particle-attached bacteria should be trapped in aerosol samples of particles larger than 1 μm, and free-floating bacteria should be located among particles smaller than 1 μm.

By utilizing 8-stage Andersen cascade impactors (Andersen samplers), size-segregated aerosol samples were collected at a southwestern coastal site of Japan in the spring of 2013−2016, when the middle latitude westerly wind in the Northern Hemisphere frequently brought dust from the Asian continent to the observation site. Viable and nonviable bacteria in each sample were counted using the LIVE/DEAD BacLight bacterial viability assay to estimate bacterial concentrations (Murata and Zhang, 2013, 2016). Bacteria detected in samples of particles larger than 1.1 μm (the cutoff size of the sampler stages) were considered particle-attached bacteria, and those in the stages of particles smaller than 1.1 μm were considered free-floating bacteria. An analysis of method confidence showed that uncertainties due to the sample collection were small (Figs. S4 and S5 in the Supplement). In this study, we focus on comparisons of the quantitative results of particle-attached and free-floating bacteria in the air and the viability of these bacteria under dust and nondust conditions.

## 2 Methods

### 2.1 Sample collection and cell enumeration

Aerosol samples were collected on the platform of a building (32.324°N, 129.993°E; 15 m above ground level and 23 m above sea level) on the seaside of Amakusa Island, southwestern Japan (Fig. S1) during several observational campaigns in the spring of 2013 to 2016. Dust plumes from the Asian continent, called Asian dust, frequently pass this area in spring. There are limited fishery and agriculture activities and few anthropogenic sources of air pollutants around the area, making the site suitable for investigating airborne bacteria in the Asian continental outflow (Murata and Zhang, 2016).

Aerosol samples were collected onto 0.2 μm pore polycarbonate filters (47 mm; Merck Millipore Ltd., Cork, Ireland) with 8-stage Andersen samplers (Model AN-200; Tokyo Dylec Corp., Japan). The flow rate of the samplers was 28.3 L min$^{-1}$. Aerosol particles were collected onto 8 filters according to the particle aerodynamic diameter ranges of >11, 7.0−11, 4.7−7.0, 3.3−4.7, 2.1−3.3, 1.1−2.1, 0.65−1.1 and 0.43−0.65 μm. The collection time of one set of samples was from approximately 3 to 24 h. Details on the sample collection are given in Table 1 and Table S1and Fig. S2 in the Supplement.

Before the collection of each sample set, all stages of the sampler were cleaned carefully, and the plates for the filters were rinsed and wiped with 70% ethanol in a clean hood to avoid contamination. A blank control for each set of samples was prepared, i.e., a blank filter was set in the sampler without sample collection. After sample collection, the filters were sealed in Petri dishes and stored at −20°C until analysis.

The viable and nonviable bacterial cells (Fig. S3) on the filters were enumerated using the LIVE/DEAD BacLight bacterial viability assay with an epifluorescence microscope (EFM; Eclipse 80i, Nikon Corp., Tokyo, Japan) as described previously (Murata and Zhang, 2016, 2013;Hu et al., 2017). Bacterial cells and other particles were detached from the aerosol-loaded polycarbonate membranes (47 mm in diameter) in a phosphate-buffered saline solution (PBS, pH 7.4) by vortex shaking and ultrasonic vibration in ice bath. Then the suspension was treated with glutaraldehyde fixation and stained with the

LIVE/DEAD BacLight Bacterial Viability Kit (L13152, Invitrogen™, Molecular Probes Inc., Eugene, Oregon, US), followed by filtration on a 25 mm diameter and 0.2 μm pore black polycarbonate membrane for bacterial enumeration. An excitation wavelength range between 450 and 490 nm (blue) was utilized, and the microscope was operated at 1000× magnification. Fluorescent green and red/orange/yellow cells with spherical shape and size close to or smaller than 1 μm in diameter were counted as viable and nonviable bacteria, respectively. There are uncertainties in the bacterial cell counting caused by the LIVE/DEAD BacLight Bacterial Viability Kit because the kit could not distinguish archaea and small eukaryotes including fungi from bacteria (Berney et al., 2007). Since the abundance of archaea and fungi in air could be several (1−6) orders of magnitude less than that of bacteria (Fröhlich-Nowoisky et al., 2016;Fröhlich-Nowoisky et al., 2014;Delort et al., 2010) and the dominant size range of fungal spores is 2−10 μm (Bauer et al., 2008), the overestimation of bacteria caused by the kit we used should be less than 10% although the uncertainties could not be quantitatively evaluated. The cell concentrations in the size-segregated particles in the air were estimated based on cell counts and the sampling of air volumes following the subtraction of the blank controls. The viability of a group of bacterial cells was defined as the ratio of the viable bacterial cells to total bacterial cells. The procedure for the experimental operation and the formulations for the estimation of cell concentrations are given in the Supplement (Text S1 in the Supplement).

The collection efficiency of airborne bacterial cells with Andersen samplers was evaluated by comparing the results to those obtained by using BioSamplers (SKC Inc., Eighty-Four, PA, US) and in-line filter holders (47 mm, Millipore Corp., Billerica, MA, US). The comparison shows that the total bacterial concentration results of the Andersen sampler were generally consistent with those of the BioSamplers and the in-line filter holders (Fig. S4).

**Table 1.** Concentration and viability of total, free-floating and particle-attached bacteria. The concentration of coarse particles (>1 μm) and the ratio of particle-attached bacteria to coarse particles are also listed. The percentages of free-floating and particle-attached bacteria are given in the parentheses. The sample ID indicates the sequence number (1 to 27) of the sample, and dust condition (D, dusty; ND, nondust) and synoptic weather (Pr, prefront; Po, postfront; AA, approaching anticyclone; A, anticyclone) during the sampling period.

| Sample ID | Synoptic weather | Coarse particles ($10^5$ m$^{-3}$) | Total bacteria | | Free-floating bacteria | | Particle-attached bacteria (PAB) | | |
| --- | --- | --- | --- | --- | --- | --- | --- | --- | --- |
| | | | Concentration ($10^5$ cells m$^{-3}$) | Viability (%) | Concentration ($10^5$ cells m$^{-3}$) | Viability (%) | Concentration ($10^5$ cells m$^{-3}$) | Viability (%) | PAB/Coarse particles (%) |
| **Dusty (9)** | | | | | | | | | |
| 1D-Pr | Prefront | 41 | 7.8 | 84 | 1.7 (21) | 90 | 6.1 (79) | 82 | 15 |
| 2D-Po | Postfront | 32 | 2.3 | 77 | 0.5 (23) | 99 | 1.8 (77) | 71 | 6 |
| 3D-AA | Approaching anticyclone | 12 | 2.2 | 89 | 0.7 (30) | 91 | 1.6 (70) | 88 | 13 |
| 4D-Pr+Po | Pre-/postfront | 52 | 7.3 | 61 | 1.8 (25) | 71 | 5.4 (75) | 58 | 11 |
| 5D-AA | Approaching anticyclone | 21 | 4.7 | 63 | 0.7 (16) | 79 | 3.9 (84) | 60 | 19 |
| 10D-Po | Postfront | 16 | 2.5 | 40 | 0.6 (25) | 61 | 1.9 (75) | 33 | 11 |

| | | | | | | | | | |
|---|---|---|---|---|---|---|---|---|---|
| 17D-AA | Approaching anticyclone | 88 | 2.9 | 73 | 1.0 (36) | 99 | 1.9 (64) | 59 | 2 |
| 26D-Po | Postfront | 10 | 8.2 | 95 | 2.5 (30) | 97 | 5.7 (70) | 95 | 59 |
| 27D-AA | Approaching anticyclone | 15 | 1.9 | 87 | 0.9 (46) | 96 | 1.0 (54) | 78 | 7 |
| **Average** | | **32 ± 25** | **4.4 ± 2.6** | **74 ± 17** | **1.2 ± 0.7 (28 ± 9)** | **87 ± 14** | **3.2 ± 2.1 (72 ± 9)** | **69 ± 19** | **16 ± 17** |
| **Nondust (18)** | | | | | | | | | |
| 6ND-AA | Approaching anticyclone | 13 | 1.5 | 75 | 0.4 (27) | 88 | 1.1 (73) | 70 | 9 |
| 7ND-A | Anticyclone | 12 | 1.5 | 74 | 0.6 (39) | 82 | 0.9 (61) | 69 | 8 |
| 8ND-A+Pr | Anticyclone+prefront | 14 | 0.8 | 98 | 0.2 (31) | 99 | 0.5 (69) | 98 | 4 |
| 9ND-Pr | Prefront | 26 | 2.7 | 73 | 1.9 (71) | 84 | 0.8 (29) | 45 | 3 |
| 11ND-AA | Approaching anticyclone | 4 | 2.1 | 72 | 1.3 (64) | 85 | 0.8 (36) | 51 | 18 |
| 12ND-A | Anticyclone | 14 | 2.9 | 83 | 2.1 (73) | 96 | 0.8 (27) | 48 | 6 |
| 13ND-A | Anticyclone | 9 | 3.6 | 75 | 2.5 (70) | 86 | 1.1 (30) | 50 | 12 |
| 14ND-A | Anticyclone | 13 | 1.9 | 77 | 0.8 (42) | 99 | 1.1 (58) | 62 | 9 |
| 15ND-AA | Approaching anticyclone | 10 | 4.4 | 65 | 1.0 (24) | 61 | 3.4 (76) | 66 | 35 |
| 16ND-Po | Postfront | 16 | 2.5 | 89 | 0.9 (35) | 96 | 1.6 (65) | 85 | 10 |
| 18ND-AA | Approaching anticyclone | 15 | 2.9 | 91 | 0.5 (18) | 86 | 2.4 (82) | 92 | 16 |
| 19ND-A | Anticyclone | 9 | 1.1 | 72 | 0.4 (35) | 96 | 0.7 (65) | 59 | 7 |
| 20ND-A | Anticyclone | 10 | 1.0 | 77 | 0.4 (41) | 85 | 0.6 (59) | 72 | 6 |
| 21ND-A | Anticyclone | 13 | 1.7 | 63 | 1.0 (63) | 89 | 0.6 (37) | 18 | 5 |
| 22ND-A | Anticyclone | 8 | 1.2 | 40 | 0.5 (43) | 56 | 0.7 (57) | 28 | 9 |
| 23ND-Pr+Po | Pre-/postfront | 12 | 1.1 | 59 | 0.5 (48) | 88 | 0.6 (52) | 32 | 5 |
| 24ND-Po+A | Postfront/Anticyclone | 7 | 1.4 | 72 | 0.5 (38) | 88 | 0.8 (62) | 62 | 12 |
| 25ND-A | Anticyclone | 6 | 1.5 | 85 | 0.6 (40) | 95 | 0.9 (60) | 78 | 15 |
| **Average** | | **12 ± 5** | **2.0 ± 1.0** | **75 ± 13** | **0.9 ± 0.7 (44 ± 17)** | **87 ± 12** | **1.1 ± 0.7 (56 ± 17)** | **60 ± 22** | **10 ± 7** |
| **All (27)** | | | | | | | | | |
| **Average** | | **18 ± 18** | **2.8 ± 2.0** | **74 ± 14** | **1.0 ± 0.7 (39 ± 16)** | **87 ± 12** | **1.8 ± 1.7 (61 ± 16)** | **63 ± 21** | **12 ± 11** |

## 2.2 Separation of particle-attached and free-floating bacteria

In this study, bacteria in the samples of stages with particles larger than 1.1 μm were considered particle-attached, and bacteria in the samples of stages with particles ranging from 0.43−1.1 μm were considered free-floating. The resuspension of bacteria trapped by upper stages and falling onto lower stages during sample collection may cause uncertainties in the size distribution of bacteria-associated particles and the separation of particle-attached and free-floating bacteria.

The uncertainties in the estimation of particle-attached and free-floating bacteria were investigated in the laboratory (Text S2 in the Supplement). The fractions and concentrations of particle-attached bacteria obtained by the presented method were potentially underestimated. But the underestimation did not significantly affect the size distributions of particle-attached bacteria, and, in particular, the underestimation of the concentrations of particle-attached bacterial cells was less

than 10% on average (Fig. S5). The total bacterial concentration results of the Andersen sampler were generally consistent
with those of the in-line filter holders collecting total particles (Fig. S4). This result indicates that bacteria smaller than 0.43
µm, which are not available by the Andersen samplers in this study, were a minor fraction of the free-floating bacteria.

## 2.3 Atmospheric conditions

During the observation periods, the number concentrations of size-segregated airborne particles
(>0.3, >0.5, >1.0, >2.0, and >5.0 µm in diameter) were monitored with optical particle counters (OPC, KC-01D in 2013 and
KC-01E in 2014–2016, Rion Co., Ltd, Tokyo, Japan). In this study, fine particles are in the range of 0.3–1.0 µm, and those
larger than 1.0 µm are referred to as coarse particles. Meteorological conditions, including temperature, pressure, relative
humidity, precipitation, and wind speed and direction, were monitored with a weather transmitter (WXT520, Vaisala Inc.,
Helsinki, Finland). Airborne particle number concentrations and meteorological data during the observation periods are
summarized in Fig. S2 and Table 1.

On the basis of surface pressure and weather charts in the days before and after sample collection (Figs. S2 and S6),
the air parcels on the synoptic scales from which samples were collected were categorized into four groups: prefront,
postfront, approaching anticyclone, and anticyclone (Table 1 and S1). Details of the categorization are available in Murata
and Zhang (2016).

Dust episodes were identified by significant increases in coarse particle concentrations (>1 µm), the forecast for
Asian dust distributions in the east Asian region (http://www-cfors.nies.go.jp/~cfors/; Fig. S7), and the backward trajectory
of air masses calculated with the NOAA hybrid single-particle Lagrangian integrated trajectory (HYSPLIT) model
(http://ready.arl.noaa.gov/HYSPLIT_traj.php). During dust events, the coarse particle concentration largely increased at the
study site (Zhang et al., 2003). Dust particles were present in the postfront air and sometimes in the approaching anticyclone
air. The results of backward trajectory analysis during dusty and nondust episodes are shown in Fig. S8.

## 3 Results

### 3.1 Concentrations of airborne bacteria in segregated size ranges

The concentrations of bacterial cells, including viable and nonviable cells, generally showed a bimodal number-size
distribution during dust episodes (e.g., Fig. 1*a*, *b*, *d*, *f*). Most of the bacteria were present in particle fractions with aerodynamic
size ($D_p$) ranges larger than 2 µm (i.e., 2.1–3.3, 3.3–4.7 and 4.7–7.0 µm; Fig. S9). These sizes are larger than the size of
individual airborne bacterial cells (approximately 1 µm or smaller), indicating that the bacteria did not float individually in the
air but were combined with other particles or were agglomerates of bacterial cells, i.e., the bacteria were particle-attached. The
agglomerates of bacterial cells usually appear near emission sources, e.g., sea spray and leaf water (Lighthart, 1997), and

probably contributed a limited portion to particle-attached bacteria in this study. There were also many bacterial cells in the size ranges smaller than 1.1 µm, i.e., free-floating bacterial cells. Their concentration was comparable to or lower than the concentrations of bacteria in the larger size ranges (Figs. 1 and S9).

In contrast to dust episodes, during nondust periods, the number-size distribution of bacteria largely varied and did not show any trend with respect to weather conditions. In six cases during nondust periods (9ND-Pr, 11ND-AA, 12ND-A, 13ND-A, 14ND-A, and 21ND-A; Fig. S9), the bacteria appeared mainly in size ranges smaller than 1.1 µm and accumulated the most in the size range of 0.43–0.65 µm (e.g., Fig. 1c), indicating the predominance of free-floating bacteria. During most of the other nondust periods (6ND-AA, 7ND-A, 8ND-A+Pr, 16ND-Po, 19ND-A, 20ND-A, 22ND-A, 23ND-Pr+Po, 24ND-Po+A, and 25ND-A), the distributions of bacteria were similar to those during the dust periods, although the concentrations were much lower than or comparable to those in the dust episodes (e.g., Fig. 1e). There were two exceptional cases in nondust periods that had a mono-modal distribution, with peaks at 3.3–4.7 µm (15ND-AA) or larger than 11 µm (18ND-AA) (Fig. S9). Multiple processes including advection, deposition, local emission and local convective mixing could influence the size distributions. Unfortunately, we do not have enough case data to investigate statistically meaningful connections between the size distribution and those processes.

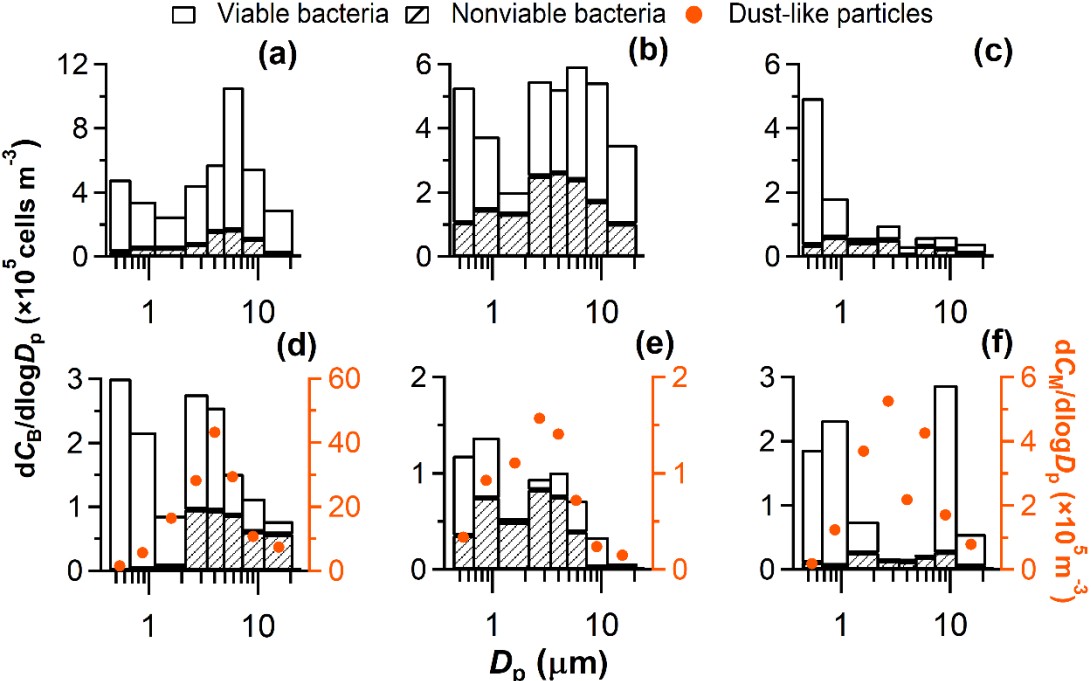

**Figure 1.** Concentrations of viable and nonviable bacteria ($C_B$) and mineral dust-like particles ($C_M$) in size-segregated airborne particles. Selected samples are shown as examples: (a) 1D-Pr; (b) 4D-Pr+Po; (c) 11ND-AA; (d) 17D-AA; (e) 22ND-A; (f) 27D-AA. The results of all sampling periods are depicted in Fig. S9 in the Supplement.

## 3.2 Concentration of particle-attached and free-floating bacteria

The report of results when data are non-normal distribution should be viewed with caution, since many statistical analyses (e.g., the average and standard deviation) are only applicable to random samples from populations with a normal distribution. Aerobiological data possibly do not have a normal distribution (Kasprzyk and Walanus, 2014;Limpert et al., 2008). Whereas, in this study, to make the comparisons among the values easily understood and avoid misunderstanding, we assume the data are normally distributed.

On average, the concentration of total bacterial cells, $4.4 \pm 2.6 \times 10^5$ cells m$^{-3}$, during dust episodes was more than twice that during nondust periods, $2.0 \pm 1.0 \times 10^5$ cells m$^{-3}$ (Table 1). This large difference (independent samples $t$ test, $p<0.05$) in concentration is consistent with the results of previous studies (Hara and Zhang, 2012;Yamaguchi et al., 2014). The concentrations of particle-attached bacterial cells during dust episodes and nondust periods were $3.2 \pm 2.1 \times 10^5$ and $1.1 \pm 0.7 \times 10^5$ cells m$^{-3}$, respectively. During dust periods particle-attached bacteria accounted for $72 \pm 9\%$ of total bacterial counts, while during nondust periods particle-attached bacteria occupied much lower proportions of $56 \pm 17\%$ (independent samples $t$ test, $p<0.05$). These results suggest that dust particles carry a substantial amount of bacterial cells on their surfaces from dust source areas to remote downstream areas.

On the other hand, the percentage of free-floating bacterial cells was in some cases higher than 70% during nondust periods (Table 1). In particular, the percentage ranged from 35% to 73% ($49 \pm 15$ % on average) under anticyclone weather conditions, when the air mass moved sluggishly and was mainly influenced by marine and local emissions and less by continental emissions (Fig. S8). Therefore, a substantial fraction of airborne bacteria were free-floating, and they were frequently the common bacteria in nondust air.

The number ratio of particle-attached bacteria to particles in the size range larger than 1.1 µm was $12 \pm 11\%$ on average (Table 1). Except for two periods when the ratios were 35% and 59%, respectively, the ratio was approximately stable ($9 \pm 5\%$ on average for the other periods), regardless of dust episodes and nondust periods (Table 1). That is, assuming that a bacteria-attached coarse particle harbors at least one bacterial cell, coarse particles including mineral dust particles with attached bacteria usually made up less than 9% of the total coarse particles. Maki et al. (2008) reported that the mineral particles with attached bacteria made up approximately 10% of the total mineral particles, with the remaining mineral particles possessing few or no bacterial cells at 800-m height above the ground in an Asian dust source region, Dunhuang, China.

The number-size distributions of bacterial cells and mineral dust-like particles (insoluble and with irregular shapes; Fig. S3) in the microscope fields of some samples were compared. In most cases, the size distributions (mode sizes) of mineral dust-like particles and bacteria in the size ranges larger than 1.1 µm showed very good consistency (Figs. 1 and S9). In some cases, the concentration of bacteria in the size ranges larger than 1.1 µm, especially nonviable bacteria, was closely correlated

with the mineral dust-like particles in the size-segregated samples (Fig. 2). These results further confirm that the bacteria observed in the large size ranges were closely associated with airborne coarse particles, i.e., they were particle attached. In some cases, the mode size ranges of the bacterial cells and the dust-like particles were inconsistent (Fig. S9), likely because the number of bacteria on the surface of each coarse particle largely varied or there were less dust-like particles in the coarse size ranges (e.g., 26D-Po). Dust-like particles were rarely observed in the size ranges smaller than 1.1 µm (Fig. S9), further indicating that the bacteria observed in those size ranges were predominantly free-floating.

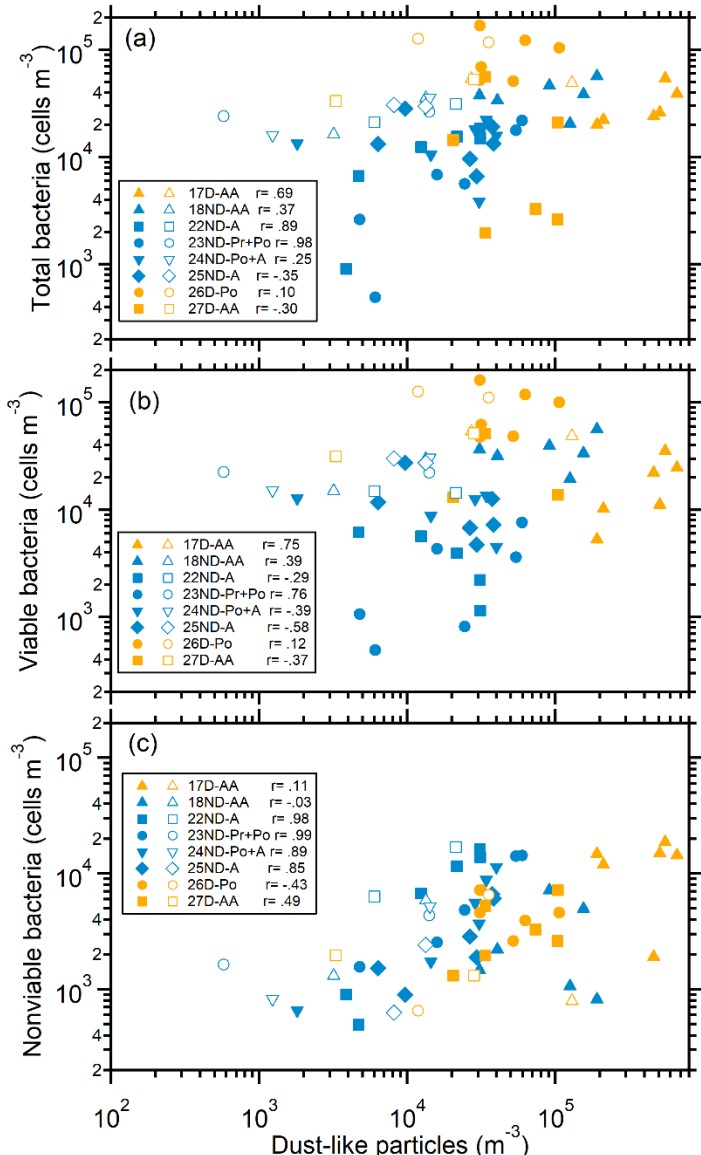

210

**Figure 2.** Relationship between bacteria and mineral dust-like particles in size-segregated aerosols. (a) Total bacteria, (b) viable bacteria, and (c) nonviable bacteria. Solid and open circles represent particles in the size ranges larger and smaller than 1.1 μm, respectively. The Pearson correlation coefficients (*r*) between bacteria and mineral dust-like particles for particles larger than 1.1 μm are shown.

### 3.3 Viabilities of particle-attached and free-floating bacteria

The viability of particle-attached bacteria varied over a wide range from 18% to 98% (63 ± 21% on average), and the viability of free-floating bacteria was between 56% and 99% (87 ± 12% on average) (Table 1), much higher than the viability of particle-attached bacteria (Paired samples $t$ test, $p$=0.00). The attachment of airborne bacteria to larger particles is expected to be favorable for retaining the viability or cultivability of cells and may indirectly increase the diversity of bacterial communities because of the possible protection of bacterial cells from harsh atmospheric conditions (Bowers et al., 2013;Prospero et al., 2005;Lighthart, 2000).

However, we found that the viability of particle-attached bacteria tended to be lower than that of free-floating bacteria, regardless of weather conditions (Table 1). This result indicates that a fraction of the particle-attached bacterial cells were either nonviable when they were blown into the air with the dust or had experienced atmospheric stressors for several days during long-distance transport and changed from a viable to a nonviable state. This is also likely the reason for the poor correlation (Pearson correlation $r$= 0.35, $p$ = 0.075) between the viability of particle-attached bacteria and the ratio of particle-attached bacteria to coarse particles (Table 1). In contrast, a large fraction of free-floating bacteria were viable. A fraction of these bacteria were likely from local areas, with a residence time (usually less than one day) shorter than that (2−3 days) of the particle-attached bacteria transported from the Asian continent (Fig. S8). The proportion of free-floating bacteria was higher under nondust conditions when the air masses moved slowly above the marine area. However, for special cases, such as the one of 20ND-A when the air was from the north due to the specific weather of west-high pressure versus east-low pressure in the westerly, a substantial fraction of the bacteria could be from the local and close areas due to the extremely strong wind. In terms of concentration, viable particle-attached bacteria were usually more abundant than viable free-floating bacteria in dust episodes (Figs. 1 and S9).

On average, the viability (74 ± 17%) of total bacteria in dusty episodes was close to the viability (75 ± 13%) of total bacteria during nondust periods (Table 1). The viability of particle-attached bacteria (69 ± 19%) during dust periods was slightly higher than that (60 ± 22%) during nondust periods. The majority of particle-attached bacteria were viable.

Free-floating bacteria exhibited a quite high viability, and the viabilities of the bacteria in dusty (87 ± 14% on average) and nondust (87 ± 12%) air were similar. The concentration of viable free-floating bacteria was $3.8 \times 10^4$–$1.5 \times 10^5$ cells m$^{-3}$, which was lower than that of particle-attached bacteria ($6.2 \times 10^4$–$5.1 \times 10^5$ cells m$^{-3}$). An increase in viable free-floating bacteria on the order of $10^5$ cell m$^{-3}$ ($1.1$–$2.2 \times 10^5$ cell m$^{-3}$) was observed when the weather was fine and the air masses moved slowly from marine areas (e.g., 9ND-Pr, 12ND-A, and 13ND-A), favoring the accumulation of bacteria emitted from local areas (Fig. S8).

## 4 Discussion

### 4.1 Implication from the comparison with literature data

There are few data on airborne bacterial cells available for comparison with the present study. Observations in the multiphase atmosphere with culture-dependent methods revealed that approximately 60−90% or even more culturable airborne bacteria were present in the size range of particles larger than 1.1 µm (Agarwal, 2017;Burrows et al., 2009;Montero et al., 2016;Raisi et al., 2013), and the median aerodynamic diameter of particles containing culturable bacteria was approximately 2–4 µm at diverse sites (Lighthart, 2000;Raisi et al., 2013;Shaffer and Lighthart, 1997;Tong and Lighthart, 2000). These results indicate the predominance of culturable particle-attached bacteria in the air, which is approximately in line with the results under dusty and nondust conditions of this study.

Early studies with single-particle analysis frequently encountered the mode size of biological aerosol particles in the size range smaller than 1 µm (Matthias-Maser et al., 1999;Matthias-Maser and Jaenicke, 1995, 2000). In contrast, recent real-time measurements using ultraviolet aerodynamic particle sizer spectrometers and wideband integrated bioaerosol sensor techniques revealed the mode size of fluorescent biological aerosol particles (FBAP) to be approximately 2–6 µm, and the particles were mainly attributed to fungal spores (Pöschl et al., 2010;Savage et al., 2017;Yue et al., 2017;Huffman et al., 2010). However, the abundant particle-attached bacteria identified in this study in size ranges larger than 2 µm indicate dust-particle-attached bacteria should not compose small fractions of real-time FBAP results in the relevant size ranges. In addition, the mode at or smaller than 1 µm observed in real-time FBAP studies is likely consistent with the presence of free-floating bacterial cells in the present study, but the comparison and discussion on the data are not confident because of the large uncertainties caused by the low counting efficiency and accuracy in submicron size ranges of the instruments used in the studies (Yue et al., 2017;Huffman et al., 2010).

Since there are rare other equivalent data for comparison, we discuss the influences of airborne bacteria according to the results obtained in this study and relevant general understandings in the following subsections.

### 4.2 Ice cloud formation

Dust particles from desert areas are constantly spread at local, regional and global scales in the atmosphere. These particles transport microorganisms across continents and oceans to remote downstream areas (Griffin, 2007;Schuerger et al., 2018). It has been shown that bacteria in the air are more effective ice nuclei at temperatures up to −2°C than abiotic particles (Ariya et al., 2009;Burrows et al., 2013;Fröhlich-Nowoisky et al., 2016;Möhler et al., 2007). Biological particles coexisting with dust particles have been detected in ice residues sampled from clouds (Creamean et al., 2013;Pratt et al., 2009), and the coexistence of dust and bacterial cells increases the ability of particles to act as ice nuclei for ice crystal formation (Tobo et al., 2019). Proteins in bacteria are ice nucleation active sites and are well protected when bacteria adhere to mineral dust surfaces (Conen et al., 2011). The attachment of bacteria to dust particles possibly increases the number of sites for ice

nucleation and consequently the ice nucleation ability of dust particles (Boose et al., 2019;Conen et al., 2011;Augustin-Bauditz et al., 2016). The present results show that up to one-tenth or more dust particles could be bacteria carriers, and the concentration of particle-attached bacteria, i.e., the number of bacteria-dust contact sites in dust episodes, was on average 3 times larger than that during nondust periods (Table 1). The occurrence of dust in remote downstream areas will significantly increase not only the concentration of bacterial cells but also the concentration of dust-bacteria mixture particles and the number of ice nucleation active sites. This phenomenon could provide important sources of nuclei for ice cloud formation under saturated meteorological conditions for icing, particularly in remote elevated air, where the concentrations of aerosol particles able to act as nuclei are usually very low (Creamean et al., 2013).

### 4.3 Ecosystem conservation and development

More than 60% of particle-attached bacteria and approximately 87% of free-floating bacteria in the dusty air remained viable. Airborne bacteria can multiply more easily after they settle into water (lakes, rivers and oceans) and soil surfaces than in the atmosphere. As a consequence, their dissemination via the atmosphere has the potential to alter the microbial biogeography, biogeochemistry and ecosystem services of downstream areas. Moreover, a recent study on phosphorus in aerosol particles in Asian continental outflow revealed that natural dust particles supplied higher ratios of bioavailable phosphorus than other types of particles as nutrients for the primary production in marine ecosystems, and the phosphorus was presumed to be from the biological particles in dust plumes (Shi et al., 2019). The dissemination of bacteria with dust in the air is much more efficient than that via other routes, such as rivers, because dust in the atmosphere can travel globally within two weeks (Uno et al., 2009). Therefore, the wide dispersal of atmospheric dust is an efficient link between bacterial communities in geographically isolated ecosystems. This linking function is likely the key process that constantly blurs the distinctions between closely related microbial species in distant areas. Thus, the diversities of microorganisms have a geographically weak gradient at the global scale, and are functions of habitat properties but not of historical/evolutionary factors (Fenchel and Finlay, 2004).

### 4.4 Health effects

Allergenic and toxic bacteria inhaled and deposited on the surface of upper respiratory tracts and lungs are suggested to provoke severe adverse health effects, regardless of whether the bacteria are viable, dead or cell fragments (Fröhlich-Nowoisky et al., 2016;Després et al., 2007). Dust particles carrying biological materials, including bacteria with pathogenic, allergenic, and adjuvant activity, can cause and aggravate respiratory disorders (Reinmuth-Selzle et al., 2017). The size distribution of bacteria-related particles in the air is particularly meaningful because the movement and deposition of the particles in the airways are size-dependent. Particles larger than 0.5 μm are deposited by sedimentation and impaction mainly in the head airways, and particles smaller than 0.5 μm can reach the lower airways by diffusion (Fröhlich-Nowoisky et al., 2016). According to the size distribution of the airborne bacteria-related particles in this study (Figs. 1 and S9), the deposition fraction and abundance of particle-attached bacteria are much higher than those of individual cells in both the upper and the

lower airways. Polymenakou et al. (2008) reported that a large fraction of airborne bacteria at respiratory particle sizes (< 3.3 μm) during an intense dust event were phylogenetic neighbors to human pathogens. He et al. (2012) suggested that Asian dust caused the exacerbation of pneumonia induced by *Klebsiella pneumoniae* due to the enhanced production of pro-inflammatory mediators in alveolar macrophages. Therefore, free-floating bacterial cells are likely to more easily influence the deep parts than the upper parts of respiratory airways, while the negative influence of particle-attached bacteria, particularly under dust conditions, is expected to be more serious in the upper parts than in the deep parts of respiratory airways.

## 5 Conclusions

In this study, we aimed to quantify the particle-attached and free-floating bacteria in dusty and nondust air in southwestern Japan using the fluorescent enumeration of bacterial cells in size-segregated aerosol samples. The bacteria showed bimodal number-size distributions during dust episodes, while the distributions largely varied during nondust periods. Particle-attached bacteria in dust episodes, with a concentration of $3.2 \pm 2.1 \times 10^5$ cells m$^{-3}$ on average, occupied $72 \pm 9$ % of the total bacteria. In contrast, this percentage was $56 \pm 17$ % during nondust periods, with a concentration of $1.1 \pm 0.7 \times 10^5$ cells m$^{-3}$. The results indicate that dust particles conveyed substantial numbers of bacterial cells on their surfaces. Viable particle-attached bacteria were more abundant than viable free-floating bacteria in dusty air, which is compatible with the previous results that larger particles harbor more viable and/or culturable bacteria than smaller particles.

The viability (approximately $63 \pm 21$ %) of particle-attached bacteria was much lower than that ($87 \pm 12$ %) of free-floating bacteria, likely because atmospheric stressors along with long-distance transport inhibited the survival of particle-attached bacteria and the entrainment of locally originating free-floating bacteria. High concentrations and viabilities of free-floating bacteria were observed in stagnant air, mostly under anticyclone conditions, suggesting that locally emitted bacteria accounted for the major fractions.

The present results, quantitatively showing the state of airborne bacteria in association with particles, i.e., particle-attached and free-floating bacteria, could have broad implications in the disciplines of atmospheric sciences, ecology, public health and climate. In addition, the methods used in this study are low cost and easily available but are time- and labor-intensive. Verification of the status of airborne bacteria using efficient techniques, such as *in situ* electron microscopy, and the exploration of the compositions, functions and activities of particle-attached and free-floating bacteria in the atmosphere, are necessary to deepen our understanding of the related fields.

**Data availability.** All data are available from the corresponding author upon request. Dataset for Figs.1 and 2 are given in Tables S2 and S3 in the supplement.

**Supplement.** The supplement related to this article is available online at: https://doi.org/…..

**Author contributions:** DZ and WH designed research; WH, KM, CF and SH performed research; WH, KM and DZ analyzed data and wrote the paper; HM and PF reviewed and commented on the paper.

**Competing interests.** The authors declare that they have no conflict of interest.

**Acknowledgments.** We thank Yuka Horikawa, Megumi Mukogawa and Miki Miyamoto for their assistance with sampling and analysis.

**Financial support.** This work was supported by the Japan Society for the Promotion of Science KAKENHI (JP16H02492, 17K18811), and the National Natural Science Foundation of China (41805118, 41977183).

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
