# Peer review of "Abundance and viability of particle-attached and free-floating bacteria in dusty and nondust air"

_Biogeosciences, 2020_

## Referee Comment (RC1) · Anonymous Referee #1 · 30 Apr 2020

The manuscript adresses a topic that is of interest to a range of scientific disciplines, as indicated in the Discussion sections. For their study, the authors have chosen a well-suited sampling location. They approached the topic with solid methods and patience to reveal new insights. I enjoyed reading the manuscript.

There are two issues, I would like the authors to think about and perhaps make according changes to the manuscript. The first issue concerns the reporting of data. Although it is common practice to report mean values and standard deviations, these metrics are not suitable when data is not normally distributed. Often with aerosol data, the value of the standard deviation is similarly large as that of the mean. In normally distributed data, about 68% of all values are within 1 standard deviation about the mean, 16% are larger and another 16% are smaller than that range. Taken seriously, a standard devi-
ation that is as large as the mean implies that 16% of the data has a negative value, which is impossible for particle concentration values. This problem and a solution to it are described in more detail in Limpert et al. (2008, Aerobiologia, 24:121–124, DOI 10.1007/s10453-008-9092-4).

The second issue is that bacteria can attach to all sides of a particle. When looking at a particle, one sees only about half of its surface. Therefore, one also sees only about half of the bacteria attached to its surface, except the particle is transparent. Did you consider this issue? If not, maybe the number of particle-attached bacteria should be re-calculated?

Minor issues

Line 11: 'aerosols more active'. Do you mean: 'aerosols to be more active'?

Line 13: Perhaps add that the size category is 'aerodynamic diameter'.

Line 43: change 'very scientifically interesting' to 'scientifically very interesting'

Line 47: The sentence starting with 'Whereas…' seem not to be complete.

Lines 54 and 66: Change 'in the spring of 2013−2016' to 'during spring in the years 2013 to 2016'.

Line 86-87: Replace 'results using BioSamplers' by 'the results to those obtained by using BioSamplers'

Line 89, 101: replace 'the holders' by 'the in-line filter holders'

Lines 126-127: 'indicating that the bacteria did not float individually in the air but were combined with other particles, i.e., the bacteria were particle-attached.' These particles could also have been other bacteria, i.e. bacteria may have been in clusters while airborne. This may affect the discussion (e.g. Line 213).

Line 146: Replace 'high difference' by 'large difference'

[Figure]

Line 159: 'moved stagnantly' seems to be a contradiction, perhaps 'moved little' or 'moved sluggishly'

Figure 2 visualises a lot of information and therefore takes a little while to be understood. That is o.k., but perhaps think of removing the trendlines because they distract from the overall pattern: Concentrations of total bacteria and viable bacteria in the size range below 1.1 um seem to increase less with increasing dust-like particle concentration, as compared to bacteria associated with particles larger than 1.1 um. In addition, why do the trendlines have different types of functions?

Line 195: 'with a residence time shorter than that of the particle-attached bacteria' Could you provide a rough estimate for the atmospheric residence times of bacteria for dusty and nondusty conditions?

Line 235: A fitting reference in this context is Augustin-Bauditz et al (2016, Atmos. Chem. Phys., 16, 5531–5543, 2016 www.atmos-chem-phys.net/16/5531/2016/)

---

## Referee Comment (RC2) · Anonymous Referee #2 · 18 May 2020

This paper by Hu et al. reports the abundance and viability of particle-attached and free-floating bacteria in air samples collected at a costal site in Japan in spring during dust and nondust episodes. The interest for bioaerosols (bacteria, fungi, yeasts, pollens, viruses...) is rather recent but growing every day, particularly because bioaerosols might have impacts on atmospheric processes (precipitation, chemistry, climate) and also on air quality (Human health, agriculture, environment). In this context, the paper presented here is quite important and interesting. Very few studies were conducted in the literature to measure the relative abundance of the bacteria attached or not to particles. In addition, the assessment of the viability of these bacteria is crucial to determine their potential impact. I am supportive of publishing this work in Biogeoscience after the following questions are addressed. General comments on Figures and Tables

1) Nomenclature of the samples The various samples are not identified in the same way depending on the figures or tables, sometimes identification is by numbers (1 to 27, see Figure 2, S7, S9), sometimes by dates (Figures 1, S2, S3, S4, S6, S8). As a consequence, it is very often hard to follow the results or comments in the text. I suggest to adopt always the same identification (number is the best). In addition, the type of event (dusty, non-dust) and the metrological information (Prefront, Postfront, Approaching anticyclone and Anticyclone) should also appear in the nomenclature of the samples For instance, the sample N°1 collected on the 19th of march 2013 which is Dusty and Prefont could be named 1 D-Pr, sample N°7 collected on the 28th of april 2013 which is Non-Dust and Anticyclone could be named 7ND-A. . . etc These nomenclatures should be homogenous in all the Tables and Figures. In parallel I suggest that Table S2 which contains very important results about the abundance and viability of free and attached bacteria should be moved to the main text of the manuscript (and not the Supplement). This table could be completed by the meteorological conditions (Prefront, Postfront, Approaching anticyclone and Anticyclone) and the samples named as suggested 1D-Pr, 7ND-A. . . etc. 2) Uncomplete presentation of the data

Could the authors explain why only some data on some samples are presented in many figures and not all of them (see Figures 2, S2,S6, S8).

Specific comments Sample collection and cell enumeration: p3 and Figure S3: Did you notice the presence of yeasts and fungi (spores) (»1ïA■m) during your experiment based on epifluorescence microscopy? Why did not you take them into account in your study? p4 and Figure S4: The authors note some discrepancy between the results obtains by the Andersen sampler used in this work and the two others samplers. The bacteria concentration seems to be usually under-estimation but the main problem in my opinion is that this under- estimated is not "constant", this is the case of the sampling on the 21th of march 2013 (figure S4). How do you take this factor into account in your results? Concentrations of bacteria of airborne bacteria in segregated size ranges (p 5, Figures 1 and S9) In my opinion it is quite difficult to really analyze the

data presented in Figure 1 and S9 in terms of random or bimodal distributions of the bacteria ... what is the scientific basis of this analysis? Is it based on visual inspection only? In addition, as noticed by the authors, when non-dust samples are analyzed both types of bacterial distribution are observed. Do you have any explanation about these segregations? Does it mean something linked to the physics of the system? This is not clear! I am wondering if we can really exploit these data. Could the authors comment on that? Concentration of particle-attached and free-floating bacteria P7, Table S2, Figure S8: The authors declare "In particular, the percentage ranged from 35% to 73% ($49\pm15$% on average) under anticyclone weather conditions, when the air parcels were from marine areas rather than from continental areas and moved stagnantly (Fig. S8). Therefore, there were a substantial amount of free-floating bacteria, and they were frequently the most common bacteria in non-dust air." This is quite interesting however when we look carefully at the data this not so true. For instance, in the event N°7 (anticyclonic, non-dust) compared to N°12 (anticyclonic, non-dust) free-floating bacteria account respectively for 39% and 73% of the total number of cells. However, N°7 (Figure 8) is clearly from marine origin with a slow motion of the air mass while N°12 moved quicker and has a continental origin. It is also true for other samples, so I do not think it is a general assumption. In addition two events (1 and 27) are missing in Figure S8 . Also did you analyze the biodiversity of the bacteria to justify your sentence "the most common bacteria"?

p7, Figure 2: The authors declare "The concentration of bacteria was usually closely correlated with the mineral dust-like particles in size-segregated samples (Fig. 2)". However, the data presented in Figure 2 are not so obvious when we look at the correlation coefficient r which are generally very low except for 2 cases in Figure 2a and 4 cases in Figure 2c. In addition, there are only 8 samples over a total number of 27. What are the results for the missing 19 samples? Finally, it would be very useful to have SEM images to confirm these conclusions. In my opinion this figure is over-interpreted and the text should be changed.

Viabilities of particle-attached and free-floating bacteria

P9, line 184, Tables 1 and S2: The authors declare: "The viability of particle-attached bacteria varied over a wide range from 18% to 98% (63±21% on average), and the viability of free-floating bacteria was between 56% and 99% (87±12%), higher than the viability of particle-attached bacteria". Did the authors performed statistical analyzes to compare these results?

P9, line 194, Tables 1 and S2, Figure S8: The authors declare " In contrast, a large fraction of free-floating bacteria were viable. A fraction of these bacteria were likely from local areas, with a residence time shorter than that of the particle-attached bacteria transported from the Asian continent". Although this assumption makes sense, it is less true when looking at the backward trajectories presented in Figure S8. For instance the event N°20 has a long trajectory from the Asian continent, far from the marine sampling site, while the event N°7 remains mainly over the sea , both samples present the same viability of the free bacteria (85 and 82% respectively), and also for the attached bacteria (72 and 69 % respectively). So in my opinion this reason is not so clear. Please could you modulate your conclusions.

P9, line 203 Tables 1 and S2, Figure S8: The authors declare "An increase in viable free-floating bacteria on the order of 105 cell m$-3$ was observed when the weather was fine and the air masses moved slowly from marine areas, favoring the accumulation of bacteria emitted from local areas (Fig. S8)". Again this observation is globally true but some examples contradict it: The event N°7 (marine origin, anticyclone, 0.6 105 cell m$-3$) does not present really higher concentrations of free bacteria compared to sample N°20 (Asian continent origin, anticyclone, , 0.6 105 cell m$-3$). So please could you modulate your conclusions.

Please also note the supplement to this comment:
https://www.biogeosciences-discuss.net/bg-2020-94/bg-2020-94-RC2-supplement.pdf

---

## Referee Comment (RC3) · Anonymous Referee #3 · 24 May 2020

**Summary:**

This work demonstrates the abundance and viability of different existential form of bacteria in dust and non-dust periods. The manuscript fits well to the scope of Biogeosciences and presents valuable results. Thus I recommend it to be published after the following moderate/minor comments listed below have been adequately addressed.

*Comments:*

1. It seems that the criterion for distinguishing particle-attached bacteria and free-floating bacteria is the size of ~ 1um, which requires more explanations. I would suggest the authors discuss the uncertainty of selected critical size.
2. Page 3, lines 78-81: I do not understand this method. Is the bacteria identified based on the fluorescence signals? If so, how the other interferences (such as SOA, PAH et al.) are excluded? Also, why you only count particles close or smaller than 1 um (free-floating bacteria)?
3. Figure 1: the label of y-axis is not right, please correct.
4. For the discussion part 4.2-4.4, it is important to study the influences of bioaerosols, but I would suggest the authors to discuss it with their own dataset.

---

## Author Comment (AC1) · 17 Jun 2020

The journal Biogeosciences

Dear the three referees,

We thank you a lot for your valuable and helpful comments.

We have prepared the point-to-point responses to the reviewers' comments and revised the manuscript accordingly. Here we submit the responses. We hope the revised manuscript meet the quality required for the publication in the journal.

Thank you very much!

Best regards, Daizhou Zhang

**Point-to-Point Response to Reviewers' Comments**

| Referee #1 |  |
|------------|--|
| Referee #2 |  |
| Referee #3 |  |

**Referee #1**

The manuscript addresses a topic that is of interest to a range of scientific disciplines, as indicated in the Discussion sections. For their study, the authors have chosen a well-suited sampling location. They approached the topic with solid methods and patience to reveal new insights. I enjoyed reading the manuscript.

**Response:** We thank the reviewer very much for the encouragement and valuable comments. The manuscript was revised according to the comments, and here we give point-to-point responses to the comments as follows.

1. There are two issues, I would like the authors to think about and perhaps make according changes to the manuscript. The first issue concerns the reporting of data. Although it is common practice to report mean values and standard deviations, these metrics are not suitable when data is not normally distributed. Often with aerosol data, the value of the standard deviation is similarly large as that of the mean. In normally distributed data, about 68% of all values are within 1 standard deviation about the mean, 16% are larger and another 16% are smaller than that range. Taken seriously, a standard deviation that is as large as the mean implies that 16% of the data has a negative value, which is impossible for particle concentration values. This problem and a solution to it are described in more detail in Limpert et al. (2008, Aerobiologia, 24:121–124, DOI: 10.1007/s10453-008-9092-4).

**Response:** We agree with the reviewer's opinion that the report of results when data are non-normal distribution should be viewed with caution.

The distributions of the concentration of bacterial cells and airborne particles, and the viability and proportion of free-floating and particle-attached bacterial cells in each dataset (n=27) are shown in Figures R1–R3.

As shown in Figure R1, the probability distributions of the concentration of bacterial cells and airborne particles are likely log-normally distributed. In contrast, the viabilities of total bacteria and particle-attached bacteria, and the proportion of free-floating and particle-attached bacteria in total bacteria likely exhibit normal distributions, while the viability of free-floating bacteria does not show an obvious distribution pattern (Figs. R2–R3).

The sample size range is small, in particular for the cases under dust conditions. In fact, processes that can affect the concentration of airborne particles are complicated, such as dispersion (transport), aerosolization and removal, and particularly the status of bacterial cells attaching to or isolated from other particles in the present cases. Factors influencing the variability of airborne bacterial cells are more complex because bacteria cells in general multiply in natural ecosystems and airborne bacteria suffer multiple stressors. Due to the limited available data, we currently cannot identify the processes leading to the distributions in each case and are unable to give rational interpretations for the causes of the distributions. We can use log-normal or normal distributions to selectively fit the observed ones, but we are afraid that will make the comparisons between the values difficult and easily misunderstanding.

For these reasons, we simply give average values for the results in Table 1 and did not apply the suggested solution (using the median of log-normal data and the multiplicative standard deviation) described in Limpert et al. (2008). Following the comment of Reviewer#2, we moved the original Table S2 to the main manuscript as Table 1 and show the results of each sample in the revision. Please refer to the response to Comment 1 of Reviewer #2. The descriptions relevant to these results in the whole text were modified accordingly.

Fig. R1. The probability distribution of the concentrations of coarse particles and bacterial cells. Upper row: raw data. Lower row: logarithmically transformed data. TB, total bacteria; FFB, free-floating bacteria; PAB, particle-attached bacteria.

Fig. R2. The probability distribution of the viability of bacterial cells. TB, total bacteria; FFB, free-floating bacteria; PAB, particle-attached bacteria.

---

## Editor Decision (ED1)

Dear authors

Thank you for the thorough revision of your manuscript. I have several questions and comments, which may be beneficial to further improve the readability and impact of the paper. Generally, I was wondering in how far it is appropriate to refer to bacteria instead of microbes in general. As far as I see, no clear characterization excluding archaea, or small eukaryotes including fungi, is available. The Live-Dead staining may be more useful for bacteria but – and correct me if I am wrong- does not exclude other organisms. It would be interesting to learn if other organisms could be transported through air, and a discussion if only in some sentences would be very interesting especially given the presented hypothesis of pathogeny but also regarding cross-fertilization of different geographic regions.

Another question which I have is that the paper describes how fractions are detached from the filters using vortexing and ultrasonic vibration. In how far can we be sure to not lose a significant fraction of organisms due to lysis induced by the latter method? Could this happen and therefore lead to an even more conservative result?

I am curious about dust transport in the context of global warming- what would be expected? How would this impact on bacteria transported by dust particles?

I also have several specific suggestions in order to improve the manuscript presentation:

L. 10 This sentence is confusing, it is unclear what is meant by 'widespread bacteria', it is also hard to understand what 'both types' refers to.

L. 11 'to be harmful'

L. 15 'blew' could be replaced by 'transported'

L. 16 please remove 'there'; 'averagely' is used frequently throughout the manuscript- I suggest rephrasing with the more common 'on average'

L. 17 please replace 'in the total bacteria' with 'of the total bacteria'

L. 20 I assume 'presented' would be more suitable than 'present'

L. 21 'substantial amounts of bacteria'

L. 22 'through the atmosphere'; I suggest replacing 'non-negligible' with 'important'

L. 23 I'm not sure what is meant by 'internally mixed assemblages'

L. 23 I suggest rephrasing as follows 'in cloud formation, in linking geographically isolated microbial communities, and possibly impact on human health.'

L. 27 'significant potential effect' is a contradiction in itself- would the effect be a potential one or is it a significant one?

L. 44 'are without any doubt'; this sentence, however, does not transport sensitive information, one way to solve the emptiness of the sentence would be to merge it with the following one.

L. 46 Pleas remove 'for such research'

L. 47 'have previously been investigated'

L. 48 'different survival mechanisms'- this would be clearer if phrased as 'various survival strategies'

L. 48 ff This sentence is very long and difficult to read, please consider simplifying.

L. 59 What is an Andersen cascade impactor

L. 72 'in spring 2013-2016'

Methods: I suggest moving section 2.3 to 2.1 so that it is easier to understand how atmospheric conditions were defined

L. 111/ 112: Please remove 'bacteria' after 'particle-attached' and after 'free-floating'

L.115 ff I suggest giving a short overview of the uncertainties to provide a solid basis for the statement on the potential underestimation. Like it is now, the reader doesn't have the chance to understand what this section is about. 'presented method'?

L. 120 'which were trapped'; What does 'with difficulty' tell us, how do we know this was difficult? If bacteria <0.43 µm were not recovered appropriately this would mean that a significant fraction of free-floating cells is probably missing?

L. 133 Please give a short overview on the categorization details

L. 136 Some level of detail on this model would be helpful.

L. 151 ff: This section refers to size ranges while figure 1 uses the categories 'viable bacteria, non-viable bacteria, dust-like particles'. This should be unified; it is difficult to connect the text to the figure otherwise.

L. 157/ L. 159: Both sentences start with 'There were', this could be avoided by merging those sentences.

L. 168/169 Here, the standard deviations were removed, as a result the numbers show a difference of a factor of roughly two. However, this seems misleading because if the ranges are included there is no significant difference. I am aware that one reviewer recommended to remove the ranges but given the danger of misinterpretation I would recommend including them, again.

L. 169 I somewhat disagree on the term 'large'. If a significant difference is claimed a statistic evaluation would be required.

L. 172 Please rephrase this sentence, a suggestion would be 'During dust periods particle-attached bacteria accounted for 72 % of total bacterial counts, while during non-dust periods, they were recovered in slightly lower proportions of 56%.

L. 'signify' is a somewhat strong statement which I am not convinced is supported. Rephrasing to 'sugggest' could solve the problem, here. In addition, how do we know where the particle-attached bacteria are transported to?

L. 175 'was in some cases higher'

L. 178 'Therefore, a substantial fraction of bacteria was free-floating.'

L. 189 – L. 192 I have difficulties understanding these two sentences- maybe this could be rephrased

Section 4.1: Consider re-naming in a way that the title refers better to the content

L. 253 'comparison'

L. 258 Does 'as warm as' mean 'up to'?

L. 308 'substantial numbers of bacterial cells'

---

## Author Response (AR2)

Editor of the journal Biogeosciences

Dear Prof. Carolin Löscher,

Thank you very much for your handling our manuscript and valuable and helpful comments. We have prepared the point-to-point responses to the comments and revised the manuscript accordingly. Here we submit the responses and the revised manuscript. We hope that the readability and the quality of the revised manuscript have been improved and met the requirements for the publication in the journal.

Thank you very much!

Best regards, Daizhou Zhang

**Point-to-Point Response to Editor's Comments**

**Dear authors**

Thank you for the thorough revision of your manuscript. I have several questions and comments, which may be beneficial to further improve the readability and impact of the paper. Generally, I was wondering in how far it is appropriate to refer to bacteria instead of microbes in general. As far as I see, no clear characterization excluding archaea, or small eukaryotes including fungi, is available. The Live-Dead staining may be more useful for bacteria but – and correct me if I am wrong- does not exclude other organisms. It would be interesting to learn if other organisms could be transported through air, and a discussion if only in some sentences would be very interesting especially given the presented hypothesis of pathogeny but also regarding cross-fertilization of different geographic regions.

Response: Thank you very much for your valuable comments.

According to the protocol of the LIVE/DEAD BacLight Bacterial Viability Kit, it provides two different nucleic acid stains—the SYTO 9 dye and propidium iodide—to rapidly distinguish live bacteria with intact plasma membranes from dead bacteria with compromised membranes. The kits enable to easily, reliably and quantitatively distinguish live and dead bacteria in minutes, even in a mixed population containing a range of bacterial types.

The LIVE/DEAD BacLight Bacterial Viability Kits have been widely used for the enumeration of bacteria, which have been applied to detect bacteria in samples of drinking water, sea water, aerosol, cloud water, and snow (Bauer et al., 2002; Boulos et al., 1999; Gasol et al., 1999; Hernlem and Ravva, 2007).

Although the LIVE/DEAD BacLight Bacterial Viability Kits are more selective for bacterial cells as implied by its name, the live/dead staining does not exclude other organisms, e.g., archaea, or small eukaryotes including fungal spores. The staining with the LIVE/DEAD BacLight Bacterial Viability Kit was shown to work not only with (eu)bacteria but also with archaea or eukaryotic cells, such as yeast (*Saccharomyces cerevisiae*) (Berney et al., 2007 and references therein, e.g., Leuko et al., 2004; Stocks, 2004; Zhang and Fang, 2004).

We used the LIVE/DEAD BacLight Bacterial Viability Kit to enumerate fluorescence particles with the size close to or smaller than 1  $\mu$ m and spherical shape and attributed them to bacterial cells, which is based on two reasons. For archaea, the abundance of archaea in air is much less than that of bacteria. For instance, Fröhlich-Nowoisky et al. (2014) found that the abundance of archaea in air was only between ~1 and ~10 gene copies per cubic meter of air, while that of bacteria was ~104 to ~106 in the same air samples. For fungal spores, the dominant size range of fungal spores is 2–10  $\mu$ m (Bauer et al., 2008), and the abundance of fungal spores in air could be one order of magnitude less than that of bacteria (Delort et al., 2010; Després et al., 2012). Therefore, the influence of archaea or eukaryotic cells, such as yeast on the presented results should be less than 10%. Unfortunately, it is impossible to directly distinguish these different types of microorganisms using staining enumeration only.

In the revision,

"There are uncertainties in the bacterial cell counting caused by the LIVE/DEAD BacLight Bacterial Viability Kit because the kit could not distinguish archaea and small eukaryotes including fungi from bacteria (Berney et al., 2007). Since the abundance of archaea and fungi in air could be several (1–6) orders of magnitude less than that of bacteria (Fröhlich-Nowoisky et al., 2014, 2016; Delort et al., 2010) and the dominant size range of fungal spores is 2–10 µm (Bauer et al., 2008), the overestimation of bacteria caused by the kit we used should be less than 10% although the uncertainties could not be quantitatively evaluated." was added in Line 93.

**References:**

Bauer, H., Claeys, M., Vermeylen, R., Schueller, E., Weinke, G., Berger, A., and Puxbaum, H.: Arabitol and mannitol as tracers for the quantification of airborne fungal spores, Atmos. Environ., 42, 588-593, 10.1016/j.atmosenv.2007.10.013, 2008.

Berney, M., Hammes, F., Bosshard, F., Weilenmann, H. U., and Egli, T.: Assessment and interpretation of bacterial viability by using the LIVE/DEAD BacLight Kit in combination with flow cytometry, Appl. Environ. Microbiol., 73, 3283-3290, 10.1128/AEM.02750-06, 2007.

Boulos, L., Prevost, M., Barbeau, B., Coallier, J., and Desjardins, R.: LIVE/DEAD® BacLight™: application of a new rapid staining method for direct enumeration of viable and total bacteria in drinking water, J. Microbiol. Methods, 37, 77-86, 10.1016/S0167-7012(99)00048-2, 1999.

Delort, A.-M., Vaïtilingom, M., Amato, P., Sancelme, M., Parazols, M., Mailhot, G., Laj, P., and Deguillaume, L.: A short overview of the microbial population in clouds: potential roles in atmospheric chemistry and nucleation processes, Atmos. Res., 98, 249-260, 10.1016/j.atmosres.2010.07.004, 2010.

Després, V. R., Huffman, J. A., Burrows, S. M., Hoose, C., Safatov, A. S., Buryak, G., Fröhlich-Nowoisky, J., Elbert, W., Andreae, M. O., and Pöschl, U.: Primary biological aerosol particles in the atmosphere: a review, Tellus B Chem. Phys. Meteorol., 64, 15598, 10.3402/tellusb.v64i0.15598, 2012.

Fröhlich-Nowoisky, J., Ruzene Nespoli, C., Pickersgill, D. A., Galand, P. E., Müller-Germann, I., Nunes, T., Gomes Cardoso, J., Almeida, S. M., Pio, C., Andreae, M. O., Conrad, R., Pöschl, U., and Després, V. R.: Diversity and seasonal dynamics of airborne archaea, Biogeosciences, 11, 6067-6079, 10.5194/bg-11-6067-2014, 2014.

Gasol, J. M., Zweifel, U. L., Peters, F., Fuhrman, J. A., and Hagström, Å.: Significance of size and nucleic acid content heterogeneity as measured by flow cytometry in natural planktonic bacteria, Appl. Environ. Microbiol., 65, 4475-4483, 1999.

Hernlem, B. J., and Ravva, S. V.: Application of flow cytometry and cell sorting to the bacterial analysis of environmental aerosol samples, J. Environ. Monit., 9, 1317-1322, 10.1039/b710512f, 2007.

Another question which I have is that the paper describes how fractions are detached from the

filters using vortexing and ultrasonic vibration. In how far can we be sure to not lose a significant fraction of organisms due to lysis induced by the latter method? Could this happen and therefore lead to an even more conservative result?

**Response:** For microbial analysis by microscopy, polycarbonate filters have been most commonly used for direct counts. Filters are often set in various kinds of air samplers during bioaerosol sampling. Microorganisms are washed from the surface of smooth-surface polycarbonate filters. The microorganisms in the wash solution are either cultured or re-filtered to uniformly distribute the microorganisms on the membrane filter. In the latter case, the microorganisms are stained and examined microscopically (Lighthart and Mohr, 1994; Jensen and Schafer, 1998; Eduard et al., 1990; Hernandez et al., 1999; Chen and Li, 2005; Maki et al., 2019; Li et al., 2011).

For the staining of microorganisms for microscopic enumeration, microorganisms on the filters are often resuspended in liquids (e.g., NaCl solution, PBS, Tween, and sterile/ultrapure water) by hand shaking, vortex and ultrasonic vibration or combined treatments. However, the time for hand shaking, vortex and ultrasonic vibration largely varies in different studies (Eduard et al., 1990; Chen and Li, 2005; Yahya et al., 2019; Li et al., 2011; Araya et al., 2019).

Sonication is concerned to cause cellular damages, especially for long sonication time. In previous studies, the sonication time in an ultrasound tank for aerosol-loaded filters in a solution ranges from 1–30 min (Yahya et al., 2019; Araya et al., 2019; Raghav et al., 2020). For a variety of environmental sample types, e.g., seawater and marine sediment, a sonification time of 30 s to 30 min has been applied to detached bacteria from other particles, gentler sonication for longer time intervals (Kepner and Pratt, 1994 and references therein). The cellular damages during the detachment procedure is generally caused by the consequent increase of temperature that cloud alter the sample by determining the bursting of prokaryotic cells with the increase of the sonication time (Danovaro, 2009).

In this study, bacteria were dislodged from the polycarbonate filters in a phosphate-buffered saline solution (PBS, pH 7.4) by vortex shaking and ultrasonic vibration in ice baths, and a gentle sonication was applied. Before we started to use this method, the dislodging time and operation conditions were confirmed in our laboratory experiments with BioSampler samples and in-line holder samples as the controls, following a number of published papers. The results using the mentioned conditions and operating procedures showed good consistency to the controls, indicating possible influence of sonication on the lysis of bacteria should be small. The filter-based techniques are not new and have been widely used before our laboratory experiments. Therefore, we don't think it is necessary to add these descriptions in the manuscript.

In the revision, "Bacterial cells and other particles were detached from the aerosol-loaded polycarbonate membranes (47 mm in diameter) by vortex shaking and ultrasonic vibration in a phosphate-buffered saline solution (PBS, pH 7.4)." was revised to "Bacterial cells and other particles were detached from the aerosol-loaded polycarbonate membranes (47 mm in diameter) in a phosphate-buffered saline solution (PBS, pH 7.4) by vortex shaking and ultrasonic vibration in in a phosphate-buffered saline solution (PBS, pH 7.4) by vortex shaking and ultrasonic vibration in ice bath."

I am curious about dust transport in the context of global warming- what would be expected? How would this impact on bacteria transported by dust particles?

**Response:** The mentioned questions are intriguing subjects. Currently, we have only a limit number of data in short periods and hardly connect them with dust transport in the context of global warming and vice versa. We hope in near future with our data integration we can answer these questions, at least to some extent.

I also have several specific suggestions in order to improve the manuscript presentation:

L. 10 This sentence is confusing, it is unclear what is meant by 'widespread bacteria', it is also hard to understand what 'both types' refers to.

**Response:** The sentence 'Widespread bacteria are a major proportion of bioaerosols and their coexistence with dust enables both types of aerosols to be more active in ice cloud formation and harmful to public health.' was revised into 'Airborne bacteria are widespread as a major proportion of atmospheric bioaerosols and their coexistence with dust particles enables both bacteria and dust particles to be more active in ice cloud formation and to be harmful to public health.'

L. 11 'to be harmful'

Response: revised.

L. 15 'blew' could be replaced by 'transported'

Response: 'blew' was replaced by 'transported'.

L. 16 please remove 'there'; 'averagely' is used frequently throughout the manuscript- I suggest rephrasing with the more common 'on average'

Response: 'there' was removed. The word 'averagely' was changed into 'on average'.

L. 17 please replace 'in the total bacteria' with 'of the total bacteria'

**Response: revised.**

L. 20 I assume 'presented' would be more suitable than 'present'

Response: 'present' was replaced by 'presented'.

L. 21 'substantial amounts of bacteria'

**Response: revised.**

L. 22 'through the atmosphere'; I suggest replacing 'non-negligible' with 'important'

**Response:** 'in the atmosphere' was revised into 'through the atmosphere', and 'non-negligible' was replaced with 'important' in the revision.

L. 23 I'm not sure what is meant by 'internally mixed assemblages'

**Response:** The mixing state of an aerosol particle indicates whether distinct, homogeneous entities occur within the same particle (internally mixed, such as an aggregate of different phases) or whether they are separated in the air (externally mixed) (Fig. R1) (Pósfai and Buseck, 2010).

Fig. R1. Schemes of externally mixed and internally mixed particles (Li et al., 2016).

L. 44 'are without any doubt'; this sentence, however, does not transport sensitive information, one way to solve the emptiness of the sentence would be to merge it with the following one.

**Response:** 'Quantitative data on the mutual state of airborne bacteria and dust particles in dusty air are no doubt scientifically very interesting (Schuerger et al., 2018). However, quantitative data are rare because of a lack of available and confident methods for such research, leaving unidentifiable uncertainties in both field observations and model simulations exploring the activities and roles of bacterial cells in atmospheric processes.' was revised to 'Quantitative data on the mutual state of airborne bacteria and dust particles in dusty air **are without any doubt** scientifically very interesting (Schuerger et al., 2018), **but are rare because of a lack of available and confident methods**, leaving unidentifiable uncertainties in both field observations and model simulations exploring the activities and roles of bacterial cells in atmospheric processes.'

L. 46 Pleas remove 'for such research'

Response: removed.

L. 47 'have previously been investigated'

Response: revised.

L. 48 'different survival mechanisms'- this would be clearer if phrased as 'various survival strategies'

**Response:** Here we mean that the ways for airborne bacteria to survive in the atmosphere are different from the ways for bacteria to survive in soils. The sentence was rephrased. Please refer to the response to the next comment.

L. 48 ff This sentence is very long and difficult to read, please consider simplifying.

**Response:** The original sentence was revised into "Whereas, the survival strategies, dispersal processes and size distribution of airborne bacteria should be different from those of bacteria in soils. The possible causes are that the aerosolization efficiency of soil bacteria from Earth surfaces varies according to bacterial species and soil types (Joung et al., 2017) and airborne bacteria suffer air turbulence and harsh atmospheric stressors (Hara and Zhang, 2012)."

L. 59 What is an Andersen cascade impactor

**Response:** It is an air sampler for the collection of size-segregated aerosol samples. "(**Andersen samplers**)" was added.

L. 72 'in spring 2013-2016'

**Response:** We revised "in the spring of 2013–2016" to "during spring in the years 2013 to 2016" according to the comment of Refree#1. In fact, the editor from Wiley Editing Services previously changed "in the spring of the years 2013–2016" into "in the spring of 2013–2016". We would like to keep the original version "in the spring of 2013–2016" that is a more common expression.

Methods: I suggest moving section 2.3 to 2.1 so that it is easier to understand how atmospheric

**conditions were defined**

**Response: Sect. 2.3 are auxiliary information. We prefer to keep it at the end of Sect. 2.**

L. 111/112: Please remove 'bacteria' after 'particle-attached' and after 'free-floating'

**Response: removed.**

L.115 ff I suggest giving a short overview of the uncertainties to provide a solid basis for the statement on the potential underestimation. Like it is now, the reader doesn't have the chance to understand what this section is about. 'presented method'?

**Response:** This paragraph is a short overview of the uncertainties. Since there are no available data and methods besides the present study for comparison, we do not have data to provide a solid basis for the discussion. We point out the uncertainties for awareness in any further studies on the subject.

**The phrase 'present method' was changed to 'presented method'.**

L. 120 'which were trapped'; What does 'with difficulty' tell us, how do we know this was difficult? If bacteria  $<0.43 \mu m$  were not recovered appropriately this would mean that a significant fraction of free-floating cells is probably missing?

**Response:** In this study, we summarize the results based on the enumeration results of eight-stage Andersen samplers. Particles with aerodynamic size smaller than 0.43  $\mu$ m were not available because the smallest size range of collected particles were 0.43  $\mu$ m. We used the word of 'difficulty' because 0.43  $\mu$ m is the cutoff size of 50% collection efficiency for particles with size 0.43  $\mu$ m and density 1 g cm-3, that means some particles smaller than 0.43  $\mu$ m could be trapped on the last filters but we cannot correctly quantify them. To investigate how many bacteria might be lost by the Anderson sampler, i.e., how many bacteria may pass over the last filter and are not trapped on any filters, we compared the results from Andersen sampler samples in some cases in the early-stage studies with the results from in-line filter (pore size: 0.2  $\mu$ m) holder samples. As shown in Figure S4, the total bacterial concentration results of the Andersen sampler were generally consistent with those of the in-line filter holders, indicating that bacteria smaller than 0.43  $\mu$ m were a minor fraction of the free-floating bacteria, that means only a minor fraction of free-floating cells was possibly missed.

In the revision, "This result indicates that bacteria smaller than 0.43  $\mu$ m, which were trapped with difficulty by the Andersen samplers, were a minor fraction of the free-floating bacteria." was revised into "This result indicates that bacteria smaller than 0.43  $\mu$ m, which are not available by the Andersen samplers in this study, were a minor fraction of the free-floating bacteria."

**L. 133 Please give a short overview on the categorization details**

**Response:**

The categorization details of synoptic weather are available in Murata and Zhang (2016) as follows:

As a cyclone passed the site, the surface pressure gradually decreased, and the weather became unstable due to warm and humid air from the southwest (i.e., prefront). It frequently rained as the cold front of the cyclone approached the site, and the passage of the cold front was recognized by the surface pressure minimum. After the passage of the cold front, dry and cold air from the Asian continent blew to the site (i.e., postfront). As anticyclones approached, air descended from the upper layers, and the near-surface wind gradually weakened with the increase in surface pressure (i.e., approaching anticyclone). When anticyclones covered the site, the air was stagnant, and the weather was clear (i.e., anticyclone).

The reason for this categorization is the distinctiveness of the origins of the aerosol particles in each of the group due to the movement of the air parcels. Postfront air is usually dry and its temperature is low, in comparison with other groups. It moves fast eastward or southeastward following cold fronts, which is the most rapid and efficient route for particulate matters, such as dust from northwestern China or anthropogenic particles from the northern China, to travel from the Asian continent to the observation area. Prefront air, which usually moves slower than the postfront air and whose movement direction is usually northeastward or northward, is warm and humid in comparison with the postfront air. It may also bring air pollutants from the Asian continent but the pollutants are usually from eastern China. The movement of anticyclone air is stagnant, warm and humid. Pollutants in the anticyclone air are usually dominated by local emissions, while sometimes with significant influence of long-distance-transported ones. The approaching anticyclone air is the air in the transition stage from postfront air to anticyclone air. In general, beside the influence of aerosols from the ocean, particulate matters in the postfront, approaching anticyclone, anticyclone and prefront air are, more or less, characterized by dust particles from northwestern China, mixture of dust and locally-emitted ones, and soot particles from eastern China, respectively.

This is not the main content in this study. We prefer to avoid repeating it here.

L. 136 Some level of detail on this model would be helpful.

**Response:** This model is used to calculate the backward trajectory of air masses. The technical details are available at the website.

In the revision, "NOAA/HYSPLIT" was changed to "NOAA hybrid single-particle Lagrangian integrated trajectory (HYSPLIT)".

L. 151 ff: This section refers to size ranges while figure 1 uses the categories 'viable bacteria, non-viable bacteria, dust-like particles'. This should be unified; it is difficult to connect the text to the figure otherwise.

**Response:** Figure 1 shows the concentrations of airborne viable and nonviable bacteria in segregated size ranges, and the *x*-axes of the sub-figures refer to size ranges of airborne particles.

The description about dust-like particles is in Line 188. We think it is useful to illustrate bacteria and dust-like particles in the same figures for comparison of the size distribution.

L. 157/L. 159: Both sentences start with 'There were', this could be avoided by merging those sentences.

**Response:** "There were multiple processes, e.g., advection, deposition, local emission and local convective mixing, that could influence the size distributions." was changed to "Multiple processes including advection, deposition, local emission and local convective mixing could influence the size distributions."

L. 168/169 Here, the standard deviations were removed, as a result the numbers show a difference of a factor of roughly two. However, this seems misleading because if the ranges are included there is no significant difference. I am aware that one reviewer recommended to remove the ranges but given the danger of misinterpretation I would recommend including them, again.

Response: We totally agree with this comment. In the revision, "The report of results when data are non-normal distribution should be viewed with caution, since many statistical analyses (e.g., the average and standard deviation) are only applicable to random samples from populations with a normal distribution. Aerobiological data possibly do not have a normal distribution (Kasprzyk and Walanus, 2014; Limpert et al., 2008). Whereas, in this study, to make the comparisons between the values easily understood and avoid misunderstanding, we assume the data are normally distributed." was added at the beginning of Sect. 3.2, and the standard deviations were added again.

L. 169 I somewhat disagree on the term 'large'. If a significant difference is claimed a statistic evaluation would be required.

**Response:** "(independent samples *t* test, p < 0.05)" was added.

L. 172 Please rephrase this sentence, a suggestion would be 'During dust periods particle-attached bacteria accounted for 72 % of total bacterial counts, while during non-dust periods, they were recovered in slightly lower proportions of 56%.

**Response:** The original sentence was revised to "During dust periods particle-attached bacteria accounted for  $72 \pm 9\%$  of total bacterial counts, while during nondust periods particle-attached bacteria occupied much lower proportions of  $56 \pm 17\%$  (independent samples t test, p<0.05)."

L. 'signify' is a somewhat strong statement which I am not convinced is supported. Rephrasing to 'sugggest' could solve the problem, here. In addition, how do we know where the particleattached bacteria are transported to?

Response: "signify" was changed to "suggest".

Here "remote downstream areas" are in comparison to "dust source areas". For example, from the Asian desert areas to the observation site.

"from dust source areas" was added.

L. 175 'was in some cases higher'

Response: revised.

L. 178 'Therefore, a substantial fraction of bacteria was free-floating.'

**Response:** The original sentence was revised to "Therefore, a substantial fraction of airborne bacteria were free-floating".

L. 189 - L. 192 I have difficulties understanding these two sentences- maybe this could be rephrased

**Response:** "In most cases, the distributions (mode sizes) of particles and bacteria in the size ranges larger than 1.1  $\mu$ m showed very good consistency (Figs. 1 and S9). In some cases, the concentration of bacteria in the size ranges larger than 1.1  $\mu$ m, especially nonviable bacteria, was closely correlated with the mineral dust-like particles in size-segregated samples (Fig. 2)." was revised to "In most cases, the size distributions (mode sizes) of mineral dust-like particles and bacteria in the size ranges larger than 1.1  $\mu$ m showed very good consistency (Figs. 1 and S9). In some cases, the concentration of bacteria in the size ranges larger than 1.1  $\mu$ m showed very good consistency (Figs. 1 and S9). In some cases, the concentration of bacteria in the size ranges larger than 1.1  $\mu$ m, especially nonviable bacteria, was closely correlated with the mineral dust-like particles in the size-segregated samples (Fig. 2)."

Section 4.1: Consider re-naming in a way that the title refers better to the content

**Response:** In the revision, the title was revised into "Implication from the comparison with literature data".

L. 253 'comparison'

Response: revised.

L. 258 Does 'as warm as' mean 'up to'?

Response: "as warm as" was revised to "up to".

L. 308 'substantial numbers of bacterial cells' **Response:** revised.

Thank you very much for your careful review and detailed comments.

[revised manuscript text omitted]

| Average    |                            | 18 ±18 | 2.8 ±2.0  | 74 ±14 | 1.0 ± 0.7 ( 39 ±16 ) | 87 ±12 | 1.8 ±1.7 (61 ±16 ) | 63 ±21  | 12 ±11 |
|------------|----------------------------|---------------|------------------|---------------|------------------------------------|---------------|----------------------------------|----------------|---------------|
| All (27)   |                            |               |                  |               |                                    |               |                                  |                |               |
| Average    |                            | 12 ± 5 | 2.0 ± 1.0 | 75 ±13 | 0.9 ±0.7 (44 ±17 )   | 87 ±12 | 1.1 ± 0.7 (56 ± 17 )      | 60 ±22  | 10 ±7  |
| 25ND-A     | Anticyclone                | 6             | 1.5              | 85            | 0.6 (40)                           | 95            | 0.9 (60)                         | 78             | 15            |
| 24ND-Po+A  | Postfront/Antic
yclone  | 7             | 1.4              | 72            | 0.5 (38)                           | 88            | 0.8 (62)                         | 62             | 12            |
| 23ND-Pr+Po | Pre-/postfront             | 12            | 1.1              | 59            | 0.5 (48)                           | 88            | 0.6 (52)                         | 32             | 5             |
| 22ND-A     | Anticyclone                | 8             | 1.2              | 40            | 0.5 (43)                           | 56            | 0.7 (57)                         | 28             | 9             |
| 21ND-A     | Anticyclone                | 13            | 1.7              | 63            | 1.0 (63)                           | 89            | 0.6 (37)                         | 18             | 5             |
| 20ND-A     | Anticyclone                | 10            | 1.0              | 77            | 0.4 (41)                           | 85            | 0.6 (59)                         | 72             | 6             |
| 19ND-A     | Anticyclone                | 9             | 1.1              | 72            | 0.4 (35)                           | 96            | 0.7 (65)                         | 59             | 7             |
| 18ND-AA    | Approaching anticyclone    | 15            | 2.9              | 91            | 0.5 (18)                           | 86            | 2.4 (82)                         | 92             | 16            |
| 16ND-Po    | Postfront                  | 16            | 2.5              | 89            | 0.9 (35)                           | 96            | 1.6 (65)                         | 85             | 10            |
| 15ND-AA    | Approaching
anticyclone | 10            | 4.4              | 65            | 1.0 (24)                           | 61            | 3.4 (76)                         | 66             | 35            |
| 14ND-A     | Anticyclone                | 13            | 1.9              | 77            | 0.8 (42)                           | 99            | 1.1 (58)                         | 62             | 9             |
| 13ND-A     | Anticyclone                | 9             | 3.6              | 75            | 2.5 (70)                           | 86            | 1.1 (30)                         | 50             | 12            |
| 12ND-A     | Anticyclone                | 14            | 2.9              | 83            | 2.1 (73)                           | 96            | 0.8 (27)                         | 48             | 6             |
| 11ND-AA    | Approaching                | 4             | 2.1              | 72            | 1.3 (64)                           | 85            | 0.8 (36)                         | 51             | 18            |
| 9ND-Pr     | Prefront                   | 26            | 2.7              | 73            | 1.9 (71)                           | 84            | 0.8 (29)                         | 45             | 3             |
| 8ND-A+Pr   | Anticyclone+pr             | 14            | 0.8              | 98            | 0.2 (31)                           | 99            | 0.5 (69)                         | 98             | 4             |
| 7ND-A      | Anticyclone                | 12            | 1.5              | 74            | 0.6 (39)                           | 82            | 0.9 (61)                         | 69             | 8             |
| 6ND-AA     | Approaching                | 13            | 1.5              | 75            | 0.4 (27)                           | 88            | 1.1 (73)                         | 70             | 9             |
| Average    |                            | 32 1 23       | 4.4 ±2.0  | /4 1/         | $1.2 \pm 0.7 (20 \pm 9)$           | 0/ 114 | $3.2 \pm 2.1 (72 \pm 9)$         | 09 11   | 10 ±17 |
| Avenage    | anticyclone                | 22 + 25       | 1.9              | 74 ± 17       | 1 2 ± 0 7 (28 ± 0)                 | 97 + 14       | $32 \pm 21(72 \pm 0)$            | 60 + 10 | 16 + 17       |
| 26D-Po     | Postfront                  | 10            | 8.2              | 95
87      | 2.5 (30)                           | 97            | 5.7 (70)                         | 95
78       | 59
7       |
| I /D-AA    | anticyclone                | 00            | 2.9              | /3            | 1.0 (36)                           | 99            | 1.9 (64)                         | 59             | 2             |
| 170 4 4    | A                          | 00            | 2.0              | 72            | 1.0.(26)                           | 00            | 1.0.(64)                         | 50             | 2             |

**2.2 Separation of particle-attached and free-floating bacteria**

[revised manuscript text omitted]

155 concentrations of bacteria in the larger size ranges (Figs. 1 and S9).

170

In contrast to dust episodes, during nondust periods, the number-size distribution of bacteria largely varied and did not show any trend with respect to weather conditions. In six cases during nondust periods (9ND-Pr, 11ND-AA, 12ND-A, 13ND-A, 14ND-A, and 21ND-A; Fig. S9), the bacteria appeared mainly in size ranges smaller than 1.1 µm and accumulated the most in the size range of 0.43–0.65 µm (e.g., Fig. 1*c*), indicating the predominance of free-floating bacteria. During most of the other nondust periods (6ND-AA, 7ND-A, 8ND-A+Pr, 16ND-Po, 19ND-A, 20ND-A, 22ND-A, 23ND-Pr+Po, 24ND-Po+A, and 25ND-A), the distributions of bacteria were similar to those during the dust periods, although the concentrations were much lower than or comparable to those in the dust episodes (e.g., Fig. 1*e*). There were two exceptional cases in nondust periods that had a mono-modal distribution, with peaks at 3.3–4.7 µm (15ND-AA) or larger than 11 µm (18ND-AA) (Fig. S9). Multiple processes including advection, deposition, local emission and local convective mixing could influence the size distributions. Unfortunately, we do not have enough case data to investigate statistically meaningful connections between the size distribution and those processes.